# FV-NeRV: Neural Compression for Free Viewpoint Videos

## Abstract

The delivery of free viewpoint videos (FVVs) is gaining popularity because of their ability to provide freely switchable perspectives to remote users as immersive experiences. While smooth view switching is crucial for enhancing user's experiences, FVV delivery faces a significant challenge in balancing traffic and decoding latency. The typical approach sends limited viewpoints and synthesizes the remainings on the user, reducing traffic, but increasing decoding delays. Alternatively, sending more viewpoints reduces the delay, but requires more bandwidth for transmission. In this paper, we propose a novel FVV representation format, Free Viewpoint-Neural Representation for Videos (FV-NeRV), to address this dilemma in FVV delivery. FV-NeRV reduces both traffic and decoding delay even for content with a large number of virtual viewpoints by overfitting compact neural networks to all viewpoints and pruning and quantizing the trained model. Experiments show that FV-NeRV achieves a comparable or even superior traffic reduction with a faster decoding speed compared to existing FVV codecs.

## 1 Introduction

Free-viewpoint video (FVV) Guo et al. (2024); Tanimoto et al. (2011) is an emerging technique that allows freely switchable viewing experience even with the limited number of physical cameras. For this purpose, FVV generates video frames from any desired viewpoint utilizing a limited set of texture and depth frames of multiple cameras and their positions Fehn (2004); Artois et al. (2023). This technique enables us to create a new type of immersive experience in the field of, for example, entertainment Amar et al. (2023), digital archive, and medical imaging Kitaguchi et al. (2024).

Ensuring seamless view-switching between viewpoints is vital for enhancing user experiences; thus, users naturally demand as many viewpoints as possible. However, it is not necessarily the case that all viewpoints desired by users are pre-recorded, so there arises a need to synthesize and transmit frames from viewpoints other than those actually recorded, using the frames from the recorded perspectives. Although many existing solutions focus on the generation of high quality free viewpoints Fehn (2004); Stankiewicz et al. (2018) from the limited number of physical views, we are still faced with a dilemma regarding the practical use of FVV: balancing traffic and decoding delay.

Depending on who actually handles the rendering of frames for the necessary viewpoints, we can consider sender-side rendering and user-side rendering. In the sender-side rendering, a content sender with rich computational resources synthesizes the video frames from all the viewpoints on demand in advance. We can conceal decoding delay from the users, but sending pre-rendered video frames of all the viewpoints increases traffic proportionally. In contrast, in the user-side rendering, the sender encodes video frames only from the physical viewpoints into a bitstream, distributes them to users, and commits the users to synthesize desired viewpoints locally as they want. In this way, we can avoid a rapid increase in traffic with an increase in the number of viewpoints. However, a long decoding delay caused by complex rendering operations, such as 3D warping and hole filling, prevents real-time playback, that is, limited frame per second (fps), especially with the limited computational power on the user devices.

In this paper, we propose Free Viewpoint-Neural Representation for Videos (**FV-NeRV**), a novel FVV format designed to effectively address the drawbacks of both the sender- and user-side rendering. This approach is heavily inspired by recent developments in frame-based implicit neural representation (INR), or Neural Representations for Videos (NeRV) Chen et al. (2021); Kwan et al.

(2023); Lee et al. (2023); He et al. (2023); Xu et al. (2024); Maiya et al. (2023); Xue et al. (2024); Bai et al. (2023); Yan et al. (2024). FV-NeRV operates on the principle of sender-side rendering while allowing all viewpoints to be transmitted to the user in an extremely lightweight manner, and this lightweight characteristic is achieved by leveraging INR techniques to overfit the transmitted content to a compact neural network. Experiments on the FVV dataset show that the proposed FV-NeRV simultaneously outperforms existing FVV codecs in terms of traffic requirement and decoding speed.

The contributions of our FV-NeRV are three-fold:

- While other INR-compression derivatives only consider a mapping from frame indices to the corresponding frames, FV-NeRV introduces a novel approach by incorporating both viewpoint indices and frame indices, enabling a compact neural network to overfit multiple video sequences from different viewpoints simultaneously.
- Using both multi-view and temporal coherences, FV-NeRV efficiently compacts FVVs across multiple viewpoints without a corresponding increase in model size.
- The compactness of the network reduces computational requirements, facilitating real-time FVV decoding by avoiding compute-intensive view synthesis operations.

## 2 RELATED WORK

### 2.1 FREE VIEWPOINT VIDEO CODING

Over the past decade, the first concept of free-viewpoint video coding has been designed in multi-view video coding (MVC) in H.264/Advanced Video Coding (AVC) Ho & Oh (2007). In contrast to single-view video, free-viewpoint video coding utilizes disparity information between viewpoints and extracts matching points in different viewpoints to compute pixel-domain residuals followed by frequency conversion, quantization, and entropy coding. MVC+D Chen et al. (2014), AVC-based 3D video coding (3D-AVC) Chen & Yen (2013) and High-Efficiency Video Coding (HEVC)-based 3D video coding (3D-HEVC) Tech et al. (2016) extended the concept of MVC to the multi-view plus depth (MVD) video format. MVC+D separately encodes the texture and depth frames, whereas 3D-AVC and 3D-HEVC jointly encode the texture and depth frames Zou et al. (2014). The latest video coding standard, i.e., MPEG Immersive Video (MIV) Boyce et al. (2021); Vadakital et al. (2022), is also designed for multi-view video plus depth format. MIV integrates multiple viewpoints of texture and depth frames into a single 2D video, eliminating inter-view redundancy. The integrated texture and depth 2D videos are encoded by any standard video codec. Some recent studies have designed learning-based video coding solutions to take advantage of deep neural network architectures for video compression. For example, some studies used deep convolutional networks (DCNNs) for single-view video Hu et al. (2021), stereo video Chen et al. (2022), and FVV Yang et al. (2024). Especially, FICNet in Yang et al. (2024) has designed the learning-based texture, depth, and residual encoder and decoder for FVV sequences.

All the above-mentioned solutions are supposed to carry out view synthesis on the user side, i.e., user-side rendering, to generate the desired viewpoints from the decoded texture and depth frames. In the proposed FV-NeRV, the video frames of the desired viewpoints can be reconstructed by feeding frame and viewpoint indices to the trained INR architecture without view synthesis operations.

### 2.2 IMPLICIT NEURAL REPRESENTATION FOR SIGNAL COMPRESSION

Recent INR architectures have been designed for image and video compression. The key concept is to compact a content of interest into a neural network through supervised learning to obtain an "index-to-signal" mapping. The content sender first trains a network so that the network can take an index and generate a corresponding signal of any kind, then sends the overfitted weights to a user as a lightweight representation of the content. The user performs simple feedforward operations with in-dices and corresponding features and reconstructs images. For video compression, early studies used pixel-wise INR architectures Dupont et al. (2021; 2022), but recent studies have proposed frame-wise INR architectures. Specifically, they feed the frame index and/or corresponding embeddings to the network to generate a video frame. NeRV Chen et al. (2021) is the first work on frame-wise video compression, and there are many extensions of the NeRV architecture Kwan et al. (2023); Lee

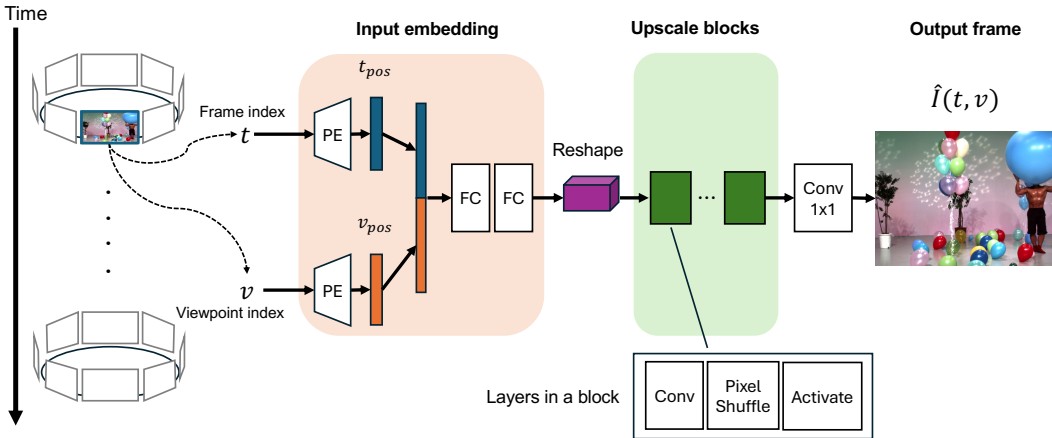

Figure 1: Overview of FV-NeRV.

et al. (2023); He et al. (2023); Xu et al. (2024); Maiya et al. (2023); Xue et al. (2024); Bai et al. (2023); Yan et al. (2024); Gomes et al. (2023); Chen et al. (2023); Zhang et al. (2024), mainly focused on improving the quality of reconstructed video frames. For example, Hierarchically-encoded NeRV (HiNeRV) Kwan et al. (2023) makes the embeddings from each video frame using ConvNext Liu et al. (2022), whereas Expedite NeRV (E-NeRV) integrates temporal frame index and spatial grid coordinates for embeddings. Boosting NeRV Zhang et al. (2024) also makes embeddings of each video frame using ConvNext and introduces a conditional decoder with sinusoidal activation function and temporal-sensitive affine transform modules for video frame reconstruction.

Our FV-NeRV is the first study of the frame-wise INR architecture for FVV sequences. Although some studies Kwan et al. (2024); Zhu et al. (2023) have designed the frame-wise INR architecture for multi-view texture and depth videos, they considered user-side rendering and needed view synthesis operations for the reconstruction of the desired viewpoints. The proposed FV-NeRV is a novel approach for sender-side rendering to simultaneously realize low-delay frame reconstruction and low data size by overfitting single and compact neural networks to various viewpoints.

## 3 FV-NERV

Fig. 1 shows the overview of the proposed FV-NeRV architecture. Let $\{I(t,v)\}_{t=1,v=1}^{T,V}$ be an FVV sequence consisting of an RGB video frame $I(t,v) \in \mathbb{R}^{H \times W \times 3}$ with $T$ frames and $V$ physical and virtual viewpoints. Here, $t, v \in [0,1]$ are normalized frame and viewpoint indices, and $H$ and $W$ are the height and width of the video frame. The proposed FV-NeRV architecture can be defined by a mapping function $f$ with learnable parameters $\boldsymbol{\theta}$ from the frame and viewpoint indices $t, v$ to the corresponding video frame $I(t,v)$ as follows:

$$f : \mathbb{R}^2 \longrightarrow \mathbb{R}^{H \times W \times 3}. \tag{1}$$

The goal of the proposed FV-NeRV architecture is to obtain the indices-to-frame mapping for the video frame of each desired viewpoint. For this purpose, we obtain the optimal function $f(t, v; \boldsymbol{\theta}) \approx f$ via network training with learnable parameters $\boldsymbol{\theta}$ using the FVV sequence $\{I(t,v)\}_{t=1,v=1}^{T,V}$.

The trained parameters are then further compressed and sent to users as $\hat{\boldsymbol{\theta}}$. Once users receive the parameters, they can reconstruct the $t$-th video frames of the desired viewpoint $v$ by feeding the corresponding indices to the FV-NeRV architecture $f(t, v; \hat{\boldsymbol{\theta}})$.

### 3.1 Model Architecture

FV-NeRV architecture $f(t, v; \boldsymbol{\theta})$ consists of a MLP and an upscaling module. The MLP part starts from positional embedding (PE) for both the frame and the viewpoint indices, which projects a

single scaler onto a high-dimensional vector. The vector maintains the positional information of the index throughout the video sequence. As proposed in Chen et al. (2021), we use a sinusoidal positional embedding with the basis $b$ and level $l$ as follows:

$$\mathbf{t}_{\text{pos}} = (\sin(b^0\pi t), \cos(b^0\pi t), \cdots, \sin(b^{l-1}\pi t), \sin(b^{l-1}\pi t)) \in \mathbb{R}^{2l},$$

$$\mathbf{v}_{\text{pos}} = (\sin(b^0\pi v), \cos(b^0\pi v), \cdots, \sin(b^{l-1}\pi v), \sin(b^{l-1}\pi v)) \in \mathbb{R}^{2l}. \quad (2)$$

These embeddings are concatenated $[\mathbf{t}_{\text{pos}}, \mathbf{v}_{\text{pos}}]$ and passed to the successive fully-connected (FC) layers. Finally, the output is reshaped to a 2D feature map $\mathbf{m} \in \mathbb{R}^{h_0 \times w_0 \times c}$, where $h_0, w_0, c$ are the height, width, and channel.

The upscaling module is made up of $L$ upscaling blocks, implemented using NeRV blocks Chen et al. (2021), which gradually enhance the resolution of the feature map. Specifically, the $l$-th block first performs 2D convolution to increase channels by $h_{l-1} \times w_{l-1} \times c \cdot s_l^2$, and then 2D pixel shuffle to increase resolution by $h_{l-1} \cdot s_l \times w_{l-1} \cdot s_l \times c$, where $s_l$ is the scaling factor for the $l$-th block. The feature map will have a resolution of $h \cdot s_1 \ldots s_L \times w \cdot s_1 \ldots s_L \times c$ after $L$ upscaling blocks, and finally, the header layer of the $1 \times 1$ 2D convolution projects the feature map to the final output with the resolution of $H \times W \times 3$ pixels.

## 3.2 Loss Function

To train the proposed FV-NeRV architecture, we integrate mean absolute error (MAE) and structural similarity (SSIM) losses as the following:

$$l = \frac{1}{T}\frac{1}{V}\sum_{t=1,v=1}^{T,V} \{\alpha \cdot \text{MAE}(f(t,v;\boldsymbol{\theta}), I(t,v)) + (1-\alpha) \cdot (1 - \text{SSIM}(f(t,v;\boldsymbol{\theta}), I(t,v)))\}, \quad (3)$$

where $\alpha$ is the hyper-parameter to balance MAE and SSIM losses. Note that MAE represents the pixel loss averaged across the whole frame. Here, the MAE loss helps minimize the pixel-level distortions, whereas the SSIM loss reduces the perceptual distortion, e.g., blockwise distortion, during the training.

## 3.3 Model Compression

We introduce model compression for the overfitted FV-NeRV model to further reduce transmission and storage costs. Like existing INR models, FV-NeRV follows model pruning, weight quantization, and weight encoding.

### 3.3.1 Model Pruning

Given the overfitted FV-NeRV model, global unstructured pruning is used to reduce the model size. Let $\boldsymbol{\theta}_q$ be the $q$-percentile value of all parameters $\boldsymbol{\theta}$. FV-NeRV sets weights with magnitudes below $\boldsymbol{\theta}_q$ to zero as follows:

$$\hat{\boldsymbol{\theta}} = \begin{cases} \boldsymbol{\theta} & \boldsymbol{\theta} \geq \boldsymbol{\theta}_q, \\ \mathbf{0} & \text{otherwise.} \end{cases} \quad (4)$$

After pruning the model, the pruned FV-NeRV parameters $\hat{\boldsymbol{\theta}}$ are fine-tuned using the same dataset.

### 3.3.2 Model Quantization and Encoding

The fine-tuned parameters are then uniformly quantized with respect to the given bit depth $N_b$, followed by the entropy coding. We employ a layer-wise quantization scheme. Given a parameter set, either weights or biases, of a layer in a FV-NeRV model $\boldsymbol{\mu} \in \hat{\boldsymbol{\theta}}$, the quantized parameter set $\boldsymbol{\mu}_q$ is given as:

$$\boldsymbol{\mu}_q = \text{round}\left(\frac{\boldsymbol{\mu} - \boldsymbol{\mu}_{\min}}{2^{N_b}}\right) * s + \boldsymbol{\mu}_{\min}, \ s = \frac{\boldsymbol{\mu}_{\max} - \boldsymbol{\mu}_{\min}}{2^{N_b}}, \quad (5)$$

where $\text{round}(\cdot)$ is a rounding function to the nearest integer and $\boldsymbol{\mu}_{\max}$ and $\boldsymbol{\mu}_{\min}$ are the maximum and minimum values in $\boldsymbol{\mu}$. The quantized tensor $\boldsymbol{\mu}_q$ is finally coded into binaries using entropy coding. FV-NeRV uses Huffman coding for binarization. Since the distribution of the tensor parameters $\boldsymbol{\mu}_q$ tends to zero, especially at small bit depths, the Huffman coding further reduces the size of the model.

# 4 EXPERIMENTS

We evaluate the performance of our FV-NeRV with respect to the quality of decoded FVV sequences and decoding delay using an FVV dataset.

## 4.1 SETTINGS

**Dataset:** We use an FVV dataset provided by Nagoya University Fujii (2013) and specifically two FVV sequences **"Balloons"** and **"Kendo"** in the dataset. This dataset provides RGB frame sequences and depth image sequences for these two sequences. For both, the frame resolution is $768 \times 1024$ pixels and the total number of frames in each viewpoint $T$ is 300.

For every FVV sequence, we choose two physical viewpoints, synthesize 9 virtual viewpoints between the physical viewpoints, and utilize these sequences of 11 viewpoints as our own dataset for the experiments. We use High-Efficiency Video Coding (HEVC) test model (HTM) software renderer Bossen et al. (2014) for virtual view synthesis using texture and depth frames. The dispersion of camera positions is $10\,\text{cm}$ for the selected physical viewpoints and $1\,\text{cm}$ for the synthesized viewpoints. When frame indices or view indices are required for frame reconstruction, we use normalized viewpoint indices $\{v | v = 0, 0.1, \cdots, 1\}$ and frame indices $\{t | t = 0, 1/300, 2/300, \cdots, 1\}$.

**Baselines:** In our experiment, we consider two scenarios for how the desired viewpoints of an FVV sequence are encoded, transmitted, and decoded: user-side rendering and sender-side rendering. We select the corresponding baseline schemes for each scenario as the competing method for our FV-NeRV.

1. **User-side rendering:** This is the typical scenario for an FVV session. The sender encodes the texture and depth image sequences for the physical viewpoints into a format and sends them to a user. The user first reconstructs the encoded texture and depth frames and performs view synthesis for all other viewpoints using the texture and depth frames. As a baseline for this scenario, we use 3D AVC test model (**3D-ATM**) Vetro et al. (2013), an existing FVV codec that generates a bitstream from given texture and depth frame sequences. 3D-ATM uses differential encoding in both the temporal and inter-view domains to account for motion and view disparity compensation.

2. **Sender-side rendering:** The sender uses the view synthesis renderer to synthesize all desired viewpoints in advance and sends all encoded video frames to a user. For this scenario, we use two types of NeRV derivatives as baselines. The first baseline is **NeRV in parallel (P-NeRV)**, where we simply train as many NeRV architectures as the number of desired viewpoints. For a total of $V$ viewpoints, P-NeRV prepares dependent networks $\{f(t; \boldsymbol{\theta}_i) : \mathbb{R} \to \mathbb{R}^{H \times W \times 3} \mid i = 1, \cdots, V\}$ and overfits each network to each view as single-view video coding. The second baseline is **NeRV in series (S-NeRV)**, where we concatenate all frames of $V$ viewpoints and treat them as a single-view sequence of length $TV$, that is, $\{t | t = 0, 1/3300, 2/3300, \cdots, 1\}$ and train them using a single NeRV architecture $f(t; \boldsymbol{\theta}) : \mathbb{R} \to \mathbb{R}^{H \times W \times 3}$ to overfit the sequence.

More detailed parameter settings for P-NeRV, S-NeRV, and FV-NeRV are listed in Appendix A.2 and A.3.

**Model size control:** We evaluate the performance of FV-NeRV and baselines under varying compression levels, and prepare four levels with different model sizes: **XS** (approximately, 4 Mbits), **S** (approximately, 8 Mbits), **M** (approximately, 12 Mbits), and **L** (approximately, 24 Mbits).

For 3D-ATM, we control different quantization parameters (QPs) for changing the size of the encoded bitstream. Here, an identical QP is used for the texture and depth video frames. Specifically,

Table 1: BD-BR values for the MS-SSIM range of 0.92 to 0.96.

| Comparison | Balloons | Kendo |
|---|---|---|
| FV-NeRV vs. 3D-ATM | -28.2 | -3.4 |
| FV-NeRV vs. P-NeRV | -92.7 | -94.1 |
| FV-NeRV vs. S-NeRV | -29.7 | -52.1 |

Table 2: Average decoding time per frame.

| Method | Decode time per frame (ms) |
|---|---|
| 3D-ATM | 230.4 |
| P/S-NeRV | 10.99 |
| **FV-NeRV** | **10.86** |

we use 33, 40, 43, and 50 QP for each compression level, respectively. For FV-NeRV and NeRV-type baselines, we control the size of the model by changing the number of network parameters and the degree of pruning and quantization. Specifically, we change the number of channels for the 2D feature map before upscaling $c_0$ to 6, 12, 26, and 58, while setting the resolution $(h_0, w_0)$ to $(12, 16)$.

**Metrics:** As the video quality metric, we use Multi-Scale Structural Similarity (MS-SSIM) Wang et al. (2003). MS-SSIM is computed for each pair of frames, yet it remains valuable for assessing the overall video quality aligned with human perception. The value ranges from 0 to 1, and a higher value close to 1 indicates a higher perceptual similarity between the original and decoded video frames. The decoding delay is measured by the average processing time required for a single frame reconstruction in the proposed FV-NeRV, P-NeRV, and S-NeRV, and for single-frame rendering in 3D-ATM.

For the rate-distortion (R-D) performance assessment, we used the Bjøntegaard delta bit-rate (BD-BR) Bjøntegaard (2001) for calculating average bit-rate differences between R-D curves for the same distortion, where negative values of BD-BR indicate the saving of the bit rate of the proposed FV-NeRV against the baseline. We use RD curves between model size and MS-SSIM index, and the MS-SSIM index was set at [0.92, 0.96] for the calculation.

### 4.2 COMPARISONS WITH BASELINES

**Rate-Distortion Performance:** Table 1 presents a summary of the calculated BD-BR savings for FV-NeRV vs. 3D-ATM, FV-NeRV vs. P-NeRV, and FV-NeRV vs. S-NeRV in each test video sequence, where negative BD-BR values indicate bitrate savings (as opposed to positive values, which indicate the bitrate required to achieve the same MS-SSIM index). It shows that the proposed FV-NeRV achieves the lowest traffic compared to the existing schemes of the sender- and user-side rendering, irrespective of the video sequences. For example, the proposed FV-NeRV realizes bitrate savings in small model sizes compared to 3D-ATM (will be mentioned in Tables 3 and 4) and this improvement brings negative BD-BR values in FV-NeRV vs. 3D-ATM. In addition, more than 90% traffic reduction can be achieved against P-NeRV for the same MS-SSIM range, since P-NeRV sends all trained weights to the desired viewpoints.

**Decoding Delay:** Table 2 shows the average decoding delay of the proposed and baseline schemes in the video sequences of "Balloons" and "Kendo". Here, the data size of the proposed FV-NeRV and existing 3D-ATM and S-NeRV is approximately 4.0 Mbits, while that of the P-NeRV is approximately 44.0 Mbits. The proposed FV-NeRV achieves the lowest decoding delay for video frame decoding, comparable to the existing P-NeRV scheme, but with a scheme with a significantly smaller data size. The 3D-ATM scheme requires view synthesis operation to reconstruct video frames of desired viewpoints, resulting in decoding delays that are more than 20 times longer than the proposed FV-NeRV.

Since P-NeRV exhibits a comparable decoding delay but has a significantly low R-D performance, we exclude it from further comparison.

**MS-SSIM Index under Different Data Sizes:** The aforementioned R-D performance evaluated the average rate savings in the given MS-SSIM range. This part considers four types of model sizes as mentioned before and compares the MS-SSIM index under the different compression levels and FVV sequences.

Tables 3 and 4 show the MS-SSIM index under the different model sizes in the proposed FV-NeRV and baseline schemes using "Balloons" and "Kendo", respectively. The evaluation results show

Table 3: MS-SSIM results of the proposed and baselines with "Balloons" video sequence under different model sizes.

| Model Size | Method | \multicolumn Average MS-SSIM for each viewpoint $v$ | | | | | | | | | | | Avg. MS-SSIM |
|---|---|---|---|---|---|---|---|---|---|---|---|---|---|
| | | 0 | 0.1 | 0.2 | 0.3 | 0.4 | 0.5 | 0.6 | 0.7 | 0.8 | 0.9 | 1.0 | |
| XS | 3D-ATM | 0.8047 | 0.8159 | 0.8239 | 0.8300 | 0.8331 | 0.8172 | 0.8360 | 0.8353 | 0.8313 | 0.8259 | 0.8158 | 0.8245 |
| | S-NeRV | 0.8034 | 0.8022 | 0.8139 | 0.8184 | 0.8196 | 0.8041 | 0.8197 | 0.8188 | 0.8154 | 0.8107 | 0.8032 | 0.8118 |
| | FV-NeRV | **0.8316** | **0.8434** | **0.8486** | **0.852** | **0.8531** | **0.8384** | **0.8526** | **0.8519** | **0.8483** | **0.8426** | **0.8347** | **0.8452** |
| S | 3D-ATM | 0.9064 | 0.9136 | 0.9188 | 0.9226 | 0.9238 | 0.9071 | 0.9241 | 0.9235 | 0.9196 | 0.9147 | 0.9075 | 0.9165 |
| | S-NeRV | 0.8858 | 0.891 | 0.898 | 0.9021 | 0.903 | 0.886 | 0.9028 | 0.9019 | 0.8978 | 0.8909 | 0.8801 | 0.8945 |
| | FV-NeRV | **0.9212** | **0.9305** | **0.9358** | **0.9392** | **0.9402** | **0.9256** | **0.94** | **0.939** | **0.9353** | **0.9298** | **0.9207** | **0.9325** |
| M | 3D-ATM | 0.9293 | 0.9353 | 0.9400 | 0.9433 | 0.9442 | 0.9274 | 0.9441 | 0.9434 | 0.9399 | 0.9354 | 0.9293 | 0.9374 |
| | S-NeRV | 0.9159 | 0.9009 | 0.923 | 0.9309 | 0.9327 | 0.9181 | 0.9351 | 0.9349 | 0.9308 | 0.9263 | 0.917 | 0.9241 |
| | FV-NeRV | **0.9479** | **0.9557** | **0.96** | **0.9628** | **0.9634** | **0.9493** | **0.9631** | **0.9624** | **0.9593** | **0.9546** | **0.9468** | **0.9568** |
| L | 3D-ATM | 0.9596 | 0.9645 | 0.9681 | 0.9707 | 0.9712 | 0.9541 | 0.9707 | 0.9702 | 0.9673 | 0.9637 | **0.9590** | 0.9654 |
| | S-NeRV | 0.944 | 0.9507 | 0.9557 | 0.9585 | 0.9586 | 0.9432 | 0.9585 | 0.9584 | 0.9556 | 0.951 | 0.9432 | 0.9525 |
| | FV-NeRV | **0.9597** | **0.9661** | **0.9698** | **0.9723** | **0.9728** | **0.9594** | **0.9725** | **0.972** | **0.9693** | **0.9653** | 0.9587 | **0.9671** |

Table 4: MS-SSIM results of the proposed and baselines with "Kendo" video sequence under different model sizes.

| Model Size | Method | \multicolumn Average MS-SSIM for each viewpoint $v$ | | | | | | | | | | | Avg. MS-SSIM |
|---|---|---|---|---|---|---|---|---|---|---|---|---|---|
| | | 0 | 0.1 | 0.2 | 0.3 | 0.4 | 0.5 | 0.6 | 0.7 | 0.8 | 0.9 | 1.0 | |
| XS | 3D-ATM | 0.8841 | 0.8927 | 0.8986 | 0.9034 | 0.9050 | 0.9062 | 0.9048 | 0.9028 | 0.8978 | 0.8916 | 0.8841 | 0.8974 |
| | S-NeRV | 0.9061 | 0.8932 | 0.899 | 0.9027 | 0.9054 | 0.8979 | 0.9064 | 0.9055 | 0.9048 | 0.9018 | 0.8934 | 0.9015 |
| | FV-NeRV | **0.9118** | **0.919** | **0.9224** | **0.9247** | **0.9253** | **0.9257** | **0.9245** | **0.9242** | **0.9214** | **0.9179** | **0.9143** | **0.9210** |
| S | 3D-ATM | 0.9360 | 0.9419 | 0.9463 | 0.9497 | 0.9505 | **0.9512** | 0.9499 | 0.9487 | 0.9445 | 0.9396 | 0.9352 | 0.9449 |
| | S-NeRV | 0.9362 | 0.922 | 0.9284 | 0.9326 | 0.9353 | 0.9276 | 0.9366 | 0.9357 | 0.9348 | 0.9317 | 0.9223 | 0.9312 |
| | FV-NeRV | **0.9386** | **0.9439** | **0.9476** | **0.95** | **0.9506** | 0.951 | **0.9501** | **0.9493** | **0.9463** | **0.9424** | **0.9376** | **0.9461** |
| M | 3D-ATM | 0.9482 | 0.9532 | 0.9572 | **0.9602** | **0.9608** | **0.9614** | **0.9602** | **0.9593** | 0.9554 | 0.9509 | **0.9478** | 0.9559 |
| | S-NeRV | 0.9473 | 0.9337 | 0.9404 | 0.9443 | 0.9467 | 0.9393 | 0.9476 | 0.9468 | 0.9461 | 0.9432 | 0.9343 | 0.9427 |
| | FV-NeRV | **0.9488** | **0.9541** | **0.9577** | 0.96 | 0.9605 | 0.9609 | 0.9601 | **0.9593** | **0.9564** | **0.9526** | 0.9476 | **0.9562** |
| L | 3D-ATM | **0.9678** | **0.9713** | **0.9743** | **0.9766** | **0.9769** | **0.9773** | **0.9761** | **0.9754** | **0.9722** | **0.9687** | **0.9670** | **0.9731** |
| | S-NeRV | 0.9613 | 0.9504 | 0.9556 | 0.9589 | 0.9609 | 0.9539 | 0.9616 | 0.9608 | 0.9601 | 0.9574 | 0.9491 | 0.9573 |
| | FV-NeRV | 0.9634 | 0.9688 | 0.9723 | 0.9745 | 0.975 | 0.9753 | 0.9745 | 0.9738 | 0.971 | 0.9673 | 0.9623 | 0.9707 |

that the proposed FV-NeRV achieves a better MS-SSIM index compared to the existing 3D-ATM, especially in the small model sizes of XS and S, although the 3D-ATM only sends the texture and depth frames of two adjacent viewpoints. We note that the reconstruction quality of 3D-ATM performs well in a large model size and this trend can be seen in existing studies between signal processing-based compression and INR-based compression Dupont et al. (2021); Chen et al. (2021).

S-NeRV suffers from low reconstruction quality in all model sizes, and thus embeddings from the viewpoint index have significant impact on the frame reconstruction of the desired viewpoints.

### 4.3 VIDEO REPRESENTATIONS IN DIFFERENT VIEWPOINTS AND MODEL SIZES

Fig. 2 shows the snapshots of the original and reconstructed "Balloons" frames of the different viewpoints in the proposed FV-NeRV and baseline schemes. Here, the frame index $t$ is 0, and the viewpoint indices $v$ are 0, 0.3, 0.7 and 1, respectively. Furthermore, the model size of the proposed and the baseline schemes is fixed to XS.

The snapshots show that the proposed FV-NeRV can reconstruct clean video frames regardless of the viewpoint positions. The reconstructed video frames in the 3D-ATM scheme are completely noisy due to large distortions in the texture and depth video frames. S-NeRV does not have an insufficient model size for frame reconstruction, and thus the shape of the tree and ground/air balloons collapses at each viewpoint.

Fig. 3 also shows the snapshots of the original and reconstructed "balloons" frames of the different model sizes in the proposed FV-NeRV and baseline schemes. Here, the frame index $t$ is 0 and the viewpoint index $v$ is 0.5. As discussed in Table 3, the proposed FV-NeRV can produce a clean video frame with a model size of S, although the other schemes still contain noise and shape distortion in the reconstructed video frames.

## 5 CONCLUSION

In this paper, we proposed FV-NeRV, a novel FVV representation that addresses the challenge of balancing traffic and decoding latency in FVV delivery. By overfitting compact neural networks to

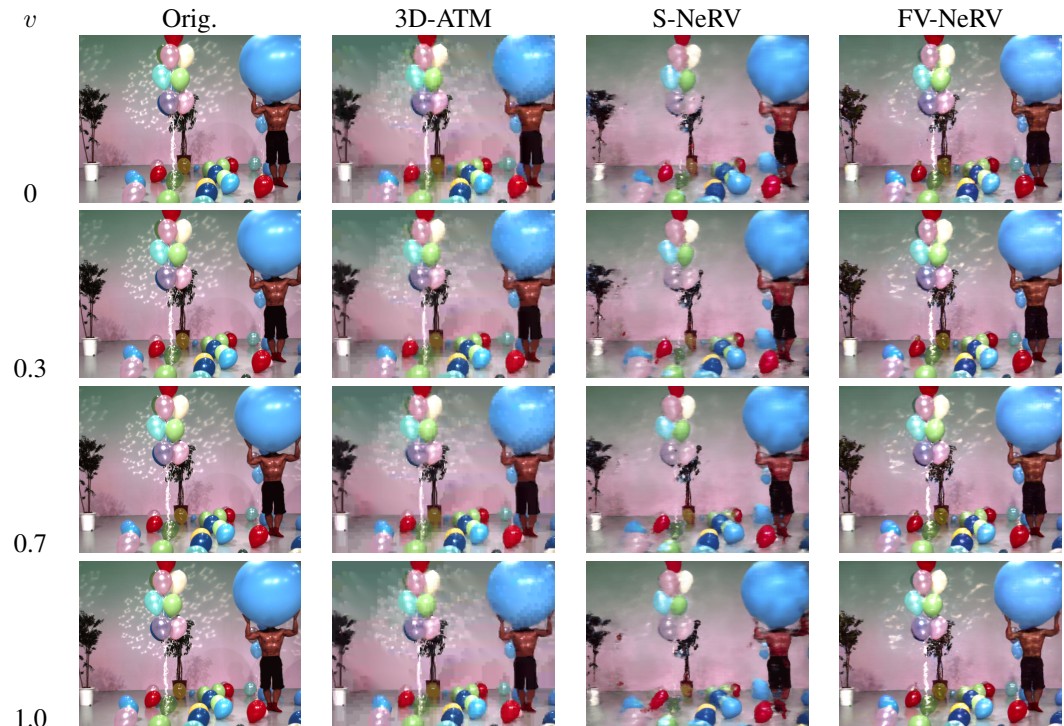

Figure 2: Snapshots of the original and reconstructed "Balloons" video frame at $t = 0$ of the XS-sized proposed FV-NeRV and baseline schemes for different viewpoint indices $v$.

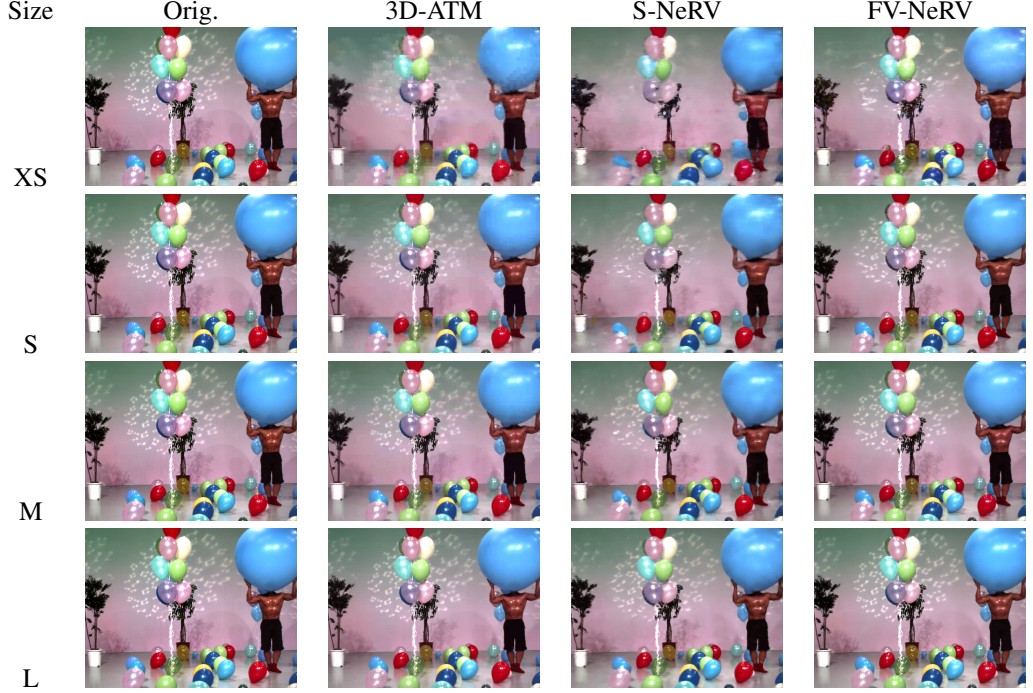

Figure 3: Snapshots of the original and reconstructed "Balloons" video frame at $t = 0$ and viewpoint index $v = 0.5$ of the proposed FV-NeRV and baseline schemes for different model sizes.

all viewpoints and reducing the model through pruning and quantization, FV-NeRV simultaneously reduces both traffic and decoding delay. Experiments demonstrated that FV-NeRV outperforms existing FVV codecs and potential NeRV derivatives, offering a more efficient solution for smooth view-switching and low data size.

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

## A APPENDIX

### A.1 VIDEO REPRESENTATIONS IN DENSE VIEWPOINTS AND DIFFERENT VIDEO SEQUENCES

We have prepared some results of video frame reconstruction as shown in Fig. 4 and 5 under the XS model size. In particular, in Fig. 5 we show the reconstructed video frames when the distance between the viewpoints is 1 cm. Although the camera arrangement is dense, the proposed FV-NeRV can reconstruct clean video frames of arbitrary viewpoints from the single compact model. In 3D-ATM, block noise occurs in the whole frame due to block-wise lossy operations for compression. The proposed FV-NeRV and the existing S-NeRV prevent block noise because the proposed upsampling blocks reconstruct the whole video frame at once. However, detailed visual information, such as the bamboo sword in each player, disappears when the size of the model is small.

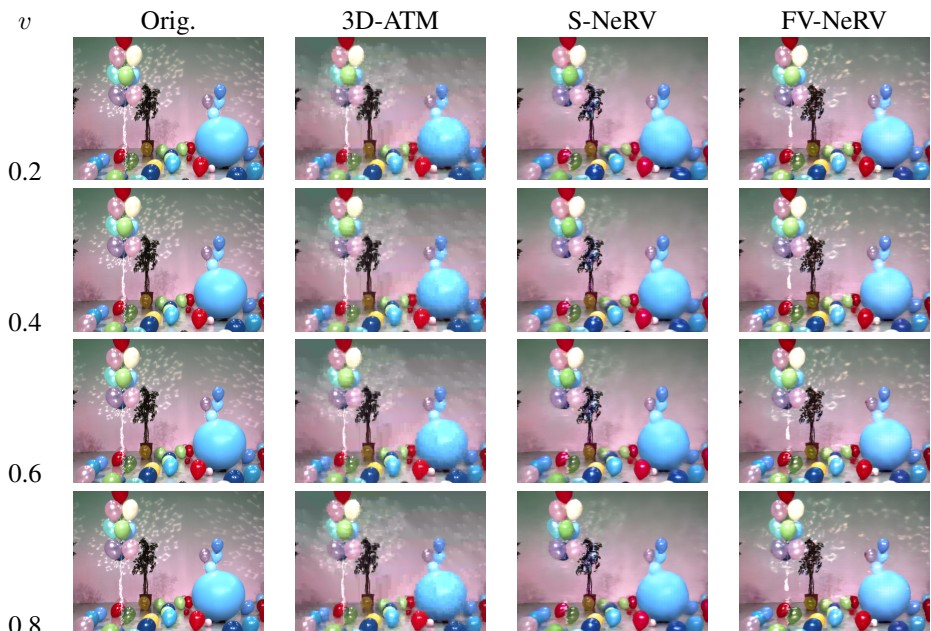

Figure 4: Reconstructed "Balloons" video frame at $t = 0.5$ of the XS-sized proposed and baseline schemes for different viewpoints $v$.

## A.2 TRAINING SETTINGS FOR NeRVs

Table 5 lists the parameter settings in P-NeRV, S-NeRV, and the proposed FV-NeRV during training, pruning, and quantization.

Table 5: Parameter settings for P-NeRV, S-NeRV, and our FV-NeRV.

|  | Name | Notation | Value |
|---|---|---|---|
| Positional Embedding | Basis | $b$ | 1.25 |
|  | Level | $l$ | 40 |
| Training parameters | Batch size | $B$ | 1 |
|  | Epoch | - | 50 |
|  | Epoch (fine-tune) | - | 50 |
|  | Optimizer | - | Adam |
|  | Learning rate | - | 5e-4 |
|  | Learning rate scheduling | - | Cosine Annealing |
|  | Loss function — weight factor | $\alpha$ | 0.7 |
| Model compression | Model pruning threshold | $q$ | 40% |
|  | Model quantization depth | $N_b$ | 8 |

## A.3 DETAIL NETWORK ARCHITECTURES FOR NeRVs

Table 6 shows the detailed network architecture of P/S/FV-NeRV. $c_0$ varies between different model sizes. SiLU stands for the activation function of Sigmoid Linear Unit that is defined as:

$$\text{SiLU}(x) = x * \sigma(x),$$

where $\sigma(x)$ is the logistic sigmoid.

Table 6: Network architecture of P/S/FV-NeRV. Here, $c_0$ varies between different model sizes.

| Group | Layer | Output Shape |
|---|---|---|
| Input |  | $[B, 1]$ (P/S-NeRV) / $[B, 2]$ (FV-NeRV) |
| Positional Embedding |  | $[B, 80]$ (P/S-NeRV) / $[B, 160]$ (FV-NeRV) |
| FC Layers | Linear | $[B, 512]$ |
|  | SiLU | - |
|  | Linear | $[B, 12 \times 16 \times c_0]$ |
|  | SiLU | - |
| Reshape |  | $[B, c_0, 12, 16]$ |
| Upsample Blocks | NeRV block 1 | $[B, c_0, 48, 64]$ |
|  | NeRV block 2 | $[B, 96, 96, 128]$ |
|  | NeRV block 3 | $[B, 96, 192, 256]$ |
|  | NeRV block 4 | $[B, 96, 384, 512]$ |
|  | NeRV block 5 | $[B, 96, 768, 1024]$ |
| Conv head |  | $[B, 3, 768, 1024]$ |
| $i$th NeRV block | 2D Convolution | $[B, c_{i-1} * s^2, h_{i-1}, w_{i-1}]$ |
|  | Pixel shuffling | $[B, c_i, h_i, w_i]$ |
|  | SiLU | - |

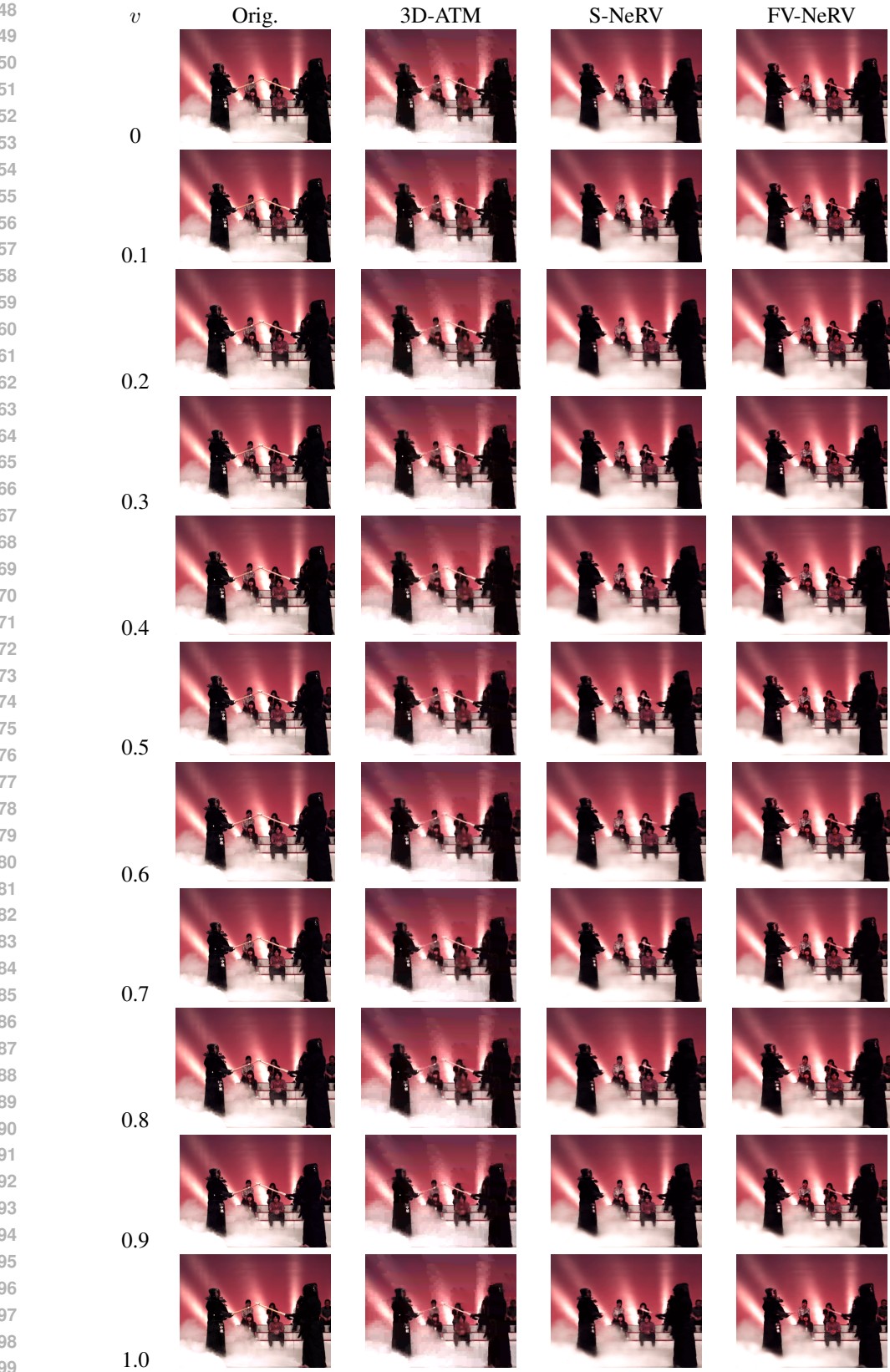

Figure 5: Reconstructed "Kendo" video frame at $t = 0.5$ of the XS-sized proposed and baseline schemes for different viewpoints $v$.

