# OpenReview forum: "FV-NeRV: Neural Compression for Free Viewpoint Videos"
_ICLR.cc/2025/Conference — ICLR 2025 Conference Withdrawn Submission_

### Official Review · Reviewer_gBko · 2024-10-22

**Soundness:** 3
**Presentation:** 3
**Contribution:** 2
**Rating:** 5
**Confidence:** 5

**Summary:**

This paper proposes a new FVV transmission method, which, compared to traditional methods, not only considers multi-viewpoints but also optimizes decoding delay and traffic.

**Strengths:**

FV-NeRV, compared to conventional methods, considers viewpoint input and can encode multi-viewpoint videos by training just one model, achieving fast decoding and excellent image quality performance with a smaller model size. At the same model size, this method achieves faster decode time per frame compared to 3D-ATM while maintaining competitive decoded image quality. Compared to S-NeRV, FV-NeRV achieves better decoded image quality with similar decode time per frame.

**Weaknesses:**

I believe this paper lacks innovation in terms of methodology, as it only adds viewpoint indexing compared to NeRV. Additionally, the experimental results are not comprehensive.

**Questions:**

1. Differences from NeRV.
In your method, I did not see significant differences from NeRV, except for the addition of the parameter "viewpoint index". The steps in Section 3.3 MODEL COMPRESSION are essentially the same. Could you explain the main differences between your method and NeRV？

2. Free Viewpoint Videos or Multi-View?
I believe your method is for multi-view rather than free viewpoint videos. The paper does not mention that your model can interpolate good new viewpoints, although this could be achieved by modifying the "viewpoint index" input. However, I did not see corresponding results, such as v = 0.55 (which I think is necessary), -0.1, and 1.1 (these two values I think are unnecessary). If possible, please supplement the results; if not, I believe the term " Free Viewpoint Videos" is not rigorous.

3. Can your model support a larger viewpoint range?
In Fig. 2, I did not see significant viewpoint changes within the range of view from 0 to 1.0. I would like to know if your method can achieve larger viewpoint range multi-view video transmission. If not, I would like to know if other existing methods can achieve this, and what the bottlenecks are. If possible, could you supplement the experimental results, or provide reasons why they cannot be supplemented?

4. Comparison with P-NeRV in Tab. 4.
I noticed that you did not provide a comparison with P-NeRV in Tab. 4. According to my understanding, P-NeRV, since it trains 11 NeRV networks for different viewpoints simultaneously, would have a relatively larger total model size, which might be the reason you did not include it in the comparison. However, if possible, could you compare P-NeRV with FV-NeRV using smaller networks for each NeRV? If possible, please supplement the experimental results; if not, please explain the reason.

If you can convince me, I will consider raising your score.

---

### Official Review · Reviewer_oarm · 2024-11-03

**Soundness:** 1
**Presentation:** 2
**Contribution:** 1
**Rating:** 1
**Confidence:** 4

**Summary:**

The paper proposed a free viewpoint video compression approach. The proposed algorithm incorporates both viewpoint indices and frame indices, enabling a compact neural network to fit multiple video sequences in FVV. It also introduced the model compression approach to reduce the model complexity.

**Strengths:**

1. The paper studied an interesting topic on FVV compression.
2. The paper is well organized and easy to follow.

**Weaknesses:**

1. The novelty of this paper is weak. The FVV compression is a well studied topic. The proposed approach simply combined the video frames and the view points in a MLP and Convolutional neural network. All these approaches are widely studied in both FVV and MVV.
2. The proposed approach is not technical sound. The authors did not throughly introduce the rational of the proposed model architecture. Typically Transformer demonstrated promising performance on sequence to sequence model. It would be necessary to study the performance of Transformer on this task.
3. The paper missed some important related work on nerual video compression, e.g. Neural 3d video synthesis from multi-view video, Neural residual radiance fields for streamably free-viewpoint videos.
4. The evaluation is also not sufficient to verify the contribution. The paper did not compare with any SOTA related work. As a video compression paper, it is necessary to evaluate the R-D and PSNR. It is also hard to tell the quality difference between the visualized images .

**Questions:**

See the weakness above.

---

### Official Review · Reviewer_WYrf · 2024-11-04

**Soundness:** 2
**Presentation:** 2
**Contribution:** 1
**Rating:** 1
**Confidence:** 5

**Summary:**

This paper proposes a novel FVV encoding method called FV-NeRV. In a fully sender-rendered setting, FV-NeRV leverages INR technology to map the frame index and viewpoint index to output frames through a neural network, utilizing model pruning and entropy coding to compress the neural network parameters.

**Strengths:**

The paper is easy to understand.

**Weaknesses:**

1. The model architecture, loss function, and model compression methods proposed in this paper are almost identical to those in NeRV, lacking any innovation.

2. Previous work has already used similar methods for compressing multi-view videos, as seen in "Zhu C, Lu G, He B, et al. Implicit-explicit Integrated Representations for Multi-view Video Compression[J]. arXiv preprint arXiv:2311.17350, 2023." Moreover, the viewpoint index proposed in this paper has already been applied in that work.

3. The paper lacks comparison methods, as it only compares INR-based methods with NeRV and two different configurations of NeRV (S-NeRV, P-NeRV). Some recent works, such as ENeRV, HNeRV, and HiNeRV, are not included in the comparisons.

4. The experimental dataset is insufficient, with only two sequences used in experiments. At the very least, the full set of sequences in the Fujii dataset mentioned in the paper should be included, and additional common FVV datasets, such as D-NeRF and MIV standard test sequences, should also be tested.

5. The quality evaluation metrics in the paper should include more indicators, such as PSNR and LPIPS, rather than only MS-SSIM.

In summary, I believe this paper is not yet ready for publication in a high-caliber conference like ICLR.

**Questions:**

1. The innovation in the method section of this paper appears too weak, as it only adds a viewpoint index to the existing NeRV framework. Are there any other improvements that were not mentioned?

2. Could you compare it with more recent INR methods, such as HiNeRV?

3. Conduct experiments on more datasets and evaluation metrics.

---

### Official Review · Reviewer_txpS · 2024-11-04

**Soundness:** 1
**Presentation:** 2
**Contribution:** 1
**Rating:** 3
**Confidence:** 5

**Summary:**

In this paper, the authors propose FV-NeRV, a neural video representation model designed to compress free-viewpoint videos. FV-NeRV can encode multiple video views into a single representation, and the authors demonstrate that it outperforms baseline models while offering fast decoding speeds.

**Strengths:**

- The proposed model addresses the free-viewpoint video coding problem, an area where deep learning-based approaches have been rarely explored.
- The proposed model demonstrates better performance than baseline models (though the baselines are relatively simple).

**Weaknesses:**

- While the authors claim that the proposed model is novel, existing works [1, 2] also address similar capabilities, despite requiring a view synthesizer only at the user side. FV-NeRV appears capable of synthesizing only a predefined set of views - the same views originally encoded, which is similar to approaches like MV-HiNeRV [1], where multiview videos are jointly encoded in a single neural representation. When novel, unseen views are unnecessary, a view synthesizer is also not needed for MV-HiNeRV.

- The model's novelty is questionable. MV-HiNeRV [1] already leverages view (feature grids are indexed by view) and patch (where the patch index can generalize frames) indices to map views. Additionally, there are no notable architectural changes compared to existing NeRV-based methods.

[1] Kwan, Ho Man, et al. "Immersive Video Compression using Implicit Neural Representations." arXiv preprint arXiv:2402.01596 (2024).

[2] Zhu, Chen, et al. "Implicit-explicit Integrated Representations for Multi-view Video Compression." arXiv preprint arXiv:2311.17350 (2023).

[3] Dziembowski, GL Adrian. "Test model 17 for MPEG immer-sive video." ISO/IEC JTC 1/SC

**Questions:**

The authors should consider the task where unseen views are required on the user side for a practicl free-viewpoint video codec, and also include the comparison to stronger baselines.

---

### Official Review · Reviewer_eTAE · 2024-11-04

**Soundness:** 1
**Presentation:** 2
**Contribution:** 1
**Rating:** 3
**Confidence:** 4

**Summary:**

The paper tackles the free-view point video compression from multi-view video inputs. In particular, the paper focused on a neural representation that can compress videos directly in the encoding stage and support multi-view render directly without heavy decoding. The method uses an implicit neural representation that first reconstruct multi-view videos via overfiting. In inference stage, it samples the multiple view indices and render the target multi-view videos. In evaluation, the method demonstrate the proposed method can achieve better compression performance compared to previous neural video encoding technique or traditional 3D video compression method.

**Strengths:**

The authors presents a good systematic analysis of traditional 3D video compression techniques and neural representation. They also presented a few good evaluation mechanisms with metrics to highlight the important factors that need to be considered.

**Weaknesses:**

The paper targets an application for free-viewpoint video compression. Compared to previous work that makes different trade-offs in the video compression rate, encoding computation, and decoding computation, the method presents a different assumption to "operates on the principle of sender-side rendering" and claims to address "the drawbacks of sender- and user-side rendering." However, it is highly unclear this assumption is sufficiently valid through the presentation in the paper which as a result leading to a weak standing for the technical contributions presented to support this claim. I will illustrate my confusions in the following main points.

1. The system. The proposed method firsts compress the full multi-video using an implicit neural network, which is evaluated as the compression rate the method achieved. During the application, it decodes the full multi-video first "on the sender side", and then claims the user does not need computation. If I understand this correctly, the authors completely ignored the compression and transmission for the multi-view videos needed from the "sender-side" to the "user-side", which makes it a different unfair comparison to traditional methods, e.g. baseline 3D-ATM. A fair comparison of the system will be to deploy the proposed method using overfitting process as the encoder computation in the sender side, and the decoder performs the full FV-NeRF inference. In this case, it will make the contrary claim to "sender-side rendering". In principle, the system can directly store the multi-view videos and transmit them, the compression is an additive stage in pre-process to encode and decode on sender-side. Transmiting the videos will rely a totally different technique.

2. The proposed system seems only considers multi-view videos but it compares 3D AVC baseline that encode texture and depth, which makes me confused why it is a meaningful comparison.

3. There are a number of factors will play an important role in the sender side rendering, including image resolution, framerate, and number of views in the video. However, the evaluation dataset only considers two video sequences with very limited number of views. I will suggest the authors to consider use more datasets to evaluate the trade-off space. The Neural 3D video dataset (CVPR 2022) can be a candidate.

4. The fixed number of decoded views are not "free-view". And the method itself also does not show any representation that can support free-view rendering, for example as Neural 3D video demonstrated. Either the authors need to adjust the system capability more precisely or need to demonstrate how the system perform in novel-view free-view video scenario.

**Questions:**

I have raised my main concerns regarding this paper in the weakness section. Among the four, I'd hope the authors can help answer the 1,2,4, in particular if I understood the system differently from the author presented. The 3rd point is a necessary experiment the authors need to consider to help evaluate the system in a revision, though I don't anticipate the authors to provide it timely in the rebuttal.

---

### Note · Authors · 2024-11-25

I have read and agree with the venue's withdrawal policy on behalf of myself and my co-authors.